# Distribution of suspended particulate matter at the equatorial transect in the Atlantic Ocean

Vadim Sivkov[1,2] and Ekaterina Bubnova[1,2]

[1]Shirshov Institute of Oceanology, Russian Academy of Sciences, 36, Nahimovskiy prospekt, Moscow, Russia, 117997
[2]Immanuel Kant Baltic Federal University, Kaliningrad, 14, A. Nevskogo str., Kaliningrad, Russia, 236016

**Correspondence:** Ekaterina Bubnova (bubnova.kat@gmail.com)

**Abstract.** A suspended particulate matter distribution against a hydrographical background was studied at the oceanographic transect across the Equatorial Atlantic in the year 2000. An area of abnormal high suspended matter volume concentrations was found above the Sierra Leone Rise in the entire water column (eastern part of the transect). The suggested explanation for the anomaly is based on the ballast hypothesis whereby solid particles are incorporated as ballast into suspended biogenic aggregates, leading to increased velocities of sinking. This occurs within the Northwest African upwelling area, where the plankton exposed to the Saharan dust abundance form a significant number of aggregates, which are lately transported equatorward via Canary Current. An intermediate nepheloid layer associated with the Deep Western Boundary Current was recorded from the South American Slope at depths of 3200–3700 m to 4300 m above the Para Abyssal Plain. Antarctic Bottom Water enriched in suspended matter was found mostly in the troughs at 40–41° W. It was detached from the bottom, coinciding with the core of the flow due to the bottom rise "dam") located up-stream. The grain size of particles along the entire transect has a polymodal distribution with 2–4 μm and 8–13 μm modes. The registered in some parts of the transect rise in percentage of the 7–21 μm-sized particles suggests the presence of the well-known coarse mode (20–60 μm) formed by aggregation of transparent exopolymer particles (mucus).

## 1 Introduction

Numerous studies have focused on characterizing the suspended particulate matter (SPM) during the past half a century. Modern understanding of the basic patterns of the SPM distribution in the ocean had emerged by the end of the 1980s (Chester, 1990). The oceanic SPM spread may be depicted as a three-layer structure with following layers (a) a surface water, (b) a clear water minimum, and (c) a deep-water (Biscaye and Eittreim, 1977; McCave, 1986; Chester, 1990). It was also indicated that the distribution of the SPM in oceans shows basically the interaction between the bottom sediments and abyssal waters movement, so SPM-rich benthic nepheloid layers (BNLs) coincide with pathways of the western boundary currents (Biscaye and Eittreim, 1977). The works of Armi (1978) and McCave (1983) pointed out the lateral advection of SPM, that occurs due to detaching of bottom mixed layers from the slope and leads to thickening and layering of BNLs. There are also strong intermediate nepheloid layers (INLs) in the clear water minimum layer in some regions (Thorpe and White, 1988).

The Geochemical Ocean Sections Study (GEOSECS) introduced an early global description of suspended particle distribution in the ocean, using mostly data on SPM collected via filtration (Craig and Turekian, 1976). The three-layer model of the SMP concentration distribution was also successively described in Brewer et al. (1976) at a transect through the western Atlantic Ocean: high concentrations were found in surface and in rapidly moving bottom waters, while low concentrations were observed in the mid-water regions of the sub-tropical gyres.

The SPM plays a crucial role in both regulation of sea water composition and material transport throughout the entire water column, experiencing various processes e.g. dissolution, decomposition, disaggregation, aggregation, etc. (Gardner et al., 1985a). It was shown by McCave (1975) that the SPM sizes follow a hyperbolic distribution with increase of fine particle abundance, whereas most of the particle flux to the bottom lies in the coarser size classes.

Sinking particles may remove the dissolved trace elements from solution (Lal, 1977) and transport them to bottom sediments. The extent to which the behavior of dissolved trace elements is dominated by suspended solids was referred as the great particle conspiracy (Turekian, 1977).

The summary in Jeandel et al. (2015) shows that fine particles (<53 μm) represent the vast majority of particles in the ocean. According to Sheldon et al. (1972) the size spectrum of small particles varies predictably both geographically and with depth. A linkage of strong BNLs and bentic storms with upper ocean dynamic was described in Gardner et al. (2018). Yet the full characterization of fine SPM distribution and composition in the ocean remains studied insufficiently due to the high research costs. Thus, research on transects of fine SPM distribution are still required.

The Equatorial Atlantic is an area of increased oceanographic interest owing to the central position on the global circulation map. The aim of this study was to explain the distribution of fine SPM concentration and size spectra on a transatlantic oceanographic transect in the equatorial zone of the Atlantic Ocean (Ioffe-2000 transect). The hydrophysical and hydrochemical data (background for the SPM distribution) have been published in sufficient detail (Sokov et al., 2002; Sarafanov et al., 2007). The SPM data, on the contrary, have been published in far less detail (Sivkov et al., 2001) due to the fact that not all the peculiarities were in accordance with general knowledge (Biscaye and Eittreim, 1977), namely, high SPM concentrations in the clear water minimum layer in the open ocean of the Eastern Atlantic, which are not INL. Recent studies of the SPM distribution and evolution allowed us to explain the features of the SPM distribution obtained on the transect.

## 2   Materials and Methods

A sublatitudinal transect (13–28 July 2000) between the continental slopes of Guinea and Brazil with 61 closely spaced stations was performed during the 8th cruise of the R/V Akademik Ioffe (Fig. 1) (Ioffe-2000). Additional surface samples of SPM were taken in the Northwest African upwelling zone (Cap Blanc area) just before the start of the main transect (10–11 July 2000). Sampling at the Ioffe-2000 transect was carried out using the CTD with rosette sampler, equipped with 5 L Niskin bottles, while the additional surface sampling was carried out with a use of plastic bucket.

The SPM volume concentration and particle size distribution were determined by the conductometric method via the Coulter counter (Zbi model) for 407 samples from the Ioffe-2000 transect and 13 samples from the Cap Blanc area (Supplement 1) (0.5

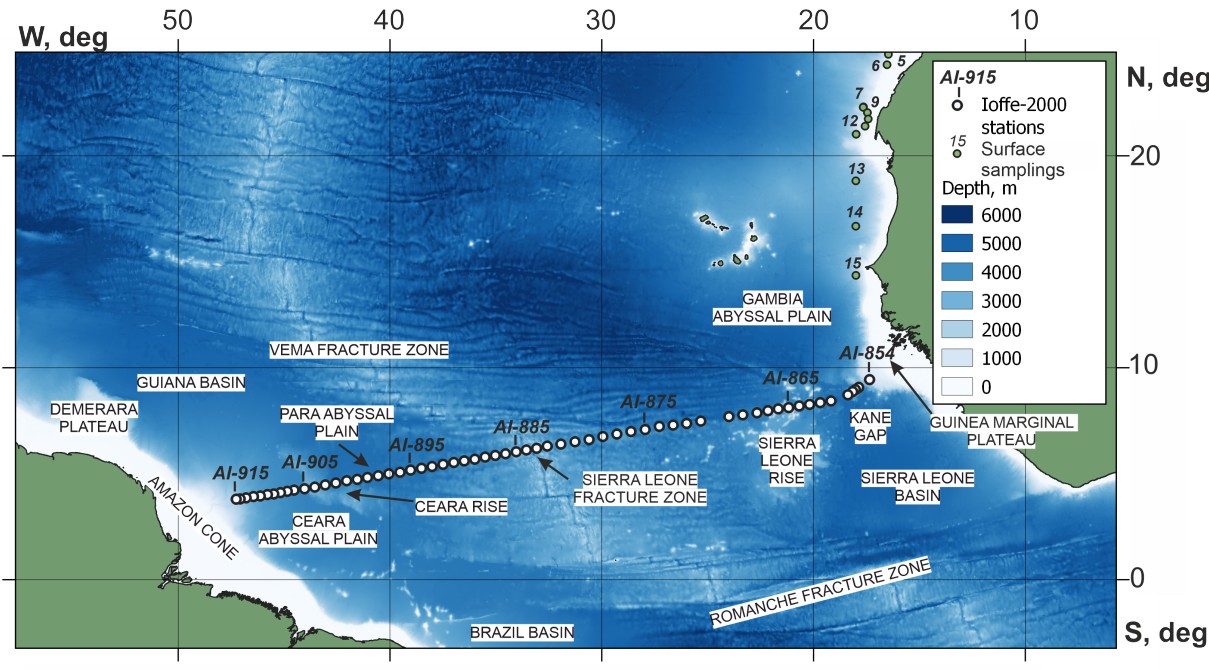

**Figure 1.** The Ioffe-2000 transect position in the study area (station numbers, every 10 stations). Bottom topography taken according to GEBCO Bathymetric Compilation Group (2020).

ml sample). The Coulter counter calibration was carried out using Coulter Electronics standard methodology using 5.96 μm diameter latex particles. The 70 μm aperture was used, which ensured counting of particles in the size range of 1.8–20.7 μm. Volume suspended solids concentration and size distribution were calculated based on the assumption of particle sphericity.

The absolute measurement error for Coulter Counter is 6% (Carder et al., 1974).The distributions of temperature, salinity, dissolved oxygen and silicates on the transect (Sokov et al., 2002; Sarafanov et al., 2007) were used in relation to the SPM data. The temperature and salinity data were recorded via an Neil Brown Mark-III CTD profile (accuracy of measurements was 0.002°C for temperature, 0.002 for salinity on the practical salinity scale). The temperature accuracy was determined via precruise and postcruise laboratory calibrations, and the salinity accuracy was calculated by comparison with the bottle data.

Water samples for additional salinity measurement and silicate concentrations were obtained using the hydrological complex mentioned above at every station. The silicate concentrations were determined according to the standard method (Mullin and Riley, 1955) (with the accuracy better than 0.2 μmol kg$^{-1}$) (Sokov et al., 2002; Sarafanov et al., 2007).

## 3 Hydrographical settings

### 3.1 Upper ocean

The upper levels of the Equatorial Atlantic water structure are dominated by the presence of large-scale westward currents and eastward countercurrents (Fig. 2A). The northern part of the studied region includes an eastward flowing North Equatorial Countercurrent (NECC) at depths 0–100 m and the North Equatorial Undercurrent (NEUC) at depths between 100 m and 300 m (Wilson et al., 1994; Bischof et al., 2003), a westward flowing North Equatorial Current (NEC) and South Equatorial Current (SEC) (Bourlès et al., 1999).

The North Brazil Current (NBC) is a major western boundary current in the Atlantic Ocean, that transports water to the north across the equator within the upper 300 m (Johns et al., 1998; Bischof et al., 2003). The North Brazil Current starts as a northern branch of water originating from the South Equatorial Current as it bifurcates nearby the Brazilian continental shelf at about 10° S, which turns the South Equatorial Current to the north, and then it merges with the North Brazil Undercurrent (NBUC). The North Brazil Current passes the equator, carrying Tropical Surface Water (TSW, surface mixed layer) at depths

of 0–100 m and the upper part of South Atlantic Central Water (SACW) (100–500 m). There is also Salinity Maximum Water on the boundary of these two water masses, that was formed in the central subtropical gyre (Stramma et al., 2005). General patterns of the western north equatorial circulation, according to Bruce et al. (1985), include a large anticyclonic eddy (diameter approximately 300–400 km) at 7–10° N and a southern eddy-like feature (3–6° N). Anticyclonic eddy features exist in the North Brazil Current upstream from the retroflection (Johns et al., 1998) and they may propagate downstream within the

North Brazil Current and serve as a catalyst for North Brazil Current ring shedding.

The North Brazil Current divides into both surface and subsurface layers of the North Equatorial Countercurrent/North Equatorial Undercurrent system (Wilson et al., 1994). There are three possible pathways for continuation of the North Brazil Current upper part (0–100 m) to the north: the coastal boundary current (Csanady, 1985; Candela et al., 1992), rings from the North Brazil Current retroflection (Didden and Schott, 1993; Richardson et al., 1994; Fratantoni et al., 1995; Johns et al., 1998), and offshore retroflection of the current into the North Equatorial Countercurrent (Mayer and Weisberg, 1993). The

North Equatorial Countercurrent lies between 3° N and 10° N, roughly considered to be the northern boundary for the South Equatorial Current (Peterson and Stramma, 1991; Bourlès et al., 1999). Both the North Equatorial Countercurrent and the South Equatorial Current are the strongest from July to September and are also at their northernmost positions during this time (Peterson and Stramma, 1991). The retroflection of the North Brazil Current is considered to be the main source of the

North Equatorial Countercurrent (Wilson et al., 1994; Bourlès et al., 1999; Schott et al., 2002), while additional sources for the latter are both the retroflected North Equatorial Current transport and the northern branch of the South Equatorial Current (Wilson et al., 1994). The thermocline North Brazil Current flow (in the 100–300-m layer) could also feed the North Equatorial Undercurrent between 2° N and 5° N (Johns et al., 1998). Some of thermocline flow of the North Equatorial Undercurrent may also recirculate back into the North Brazil Current by means of the semipermanent anticyclonic "Amazon" eddy centred near

2° N (Bruce et al., 1985).

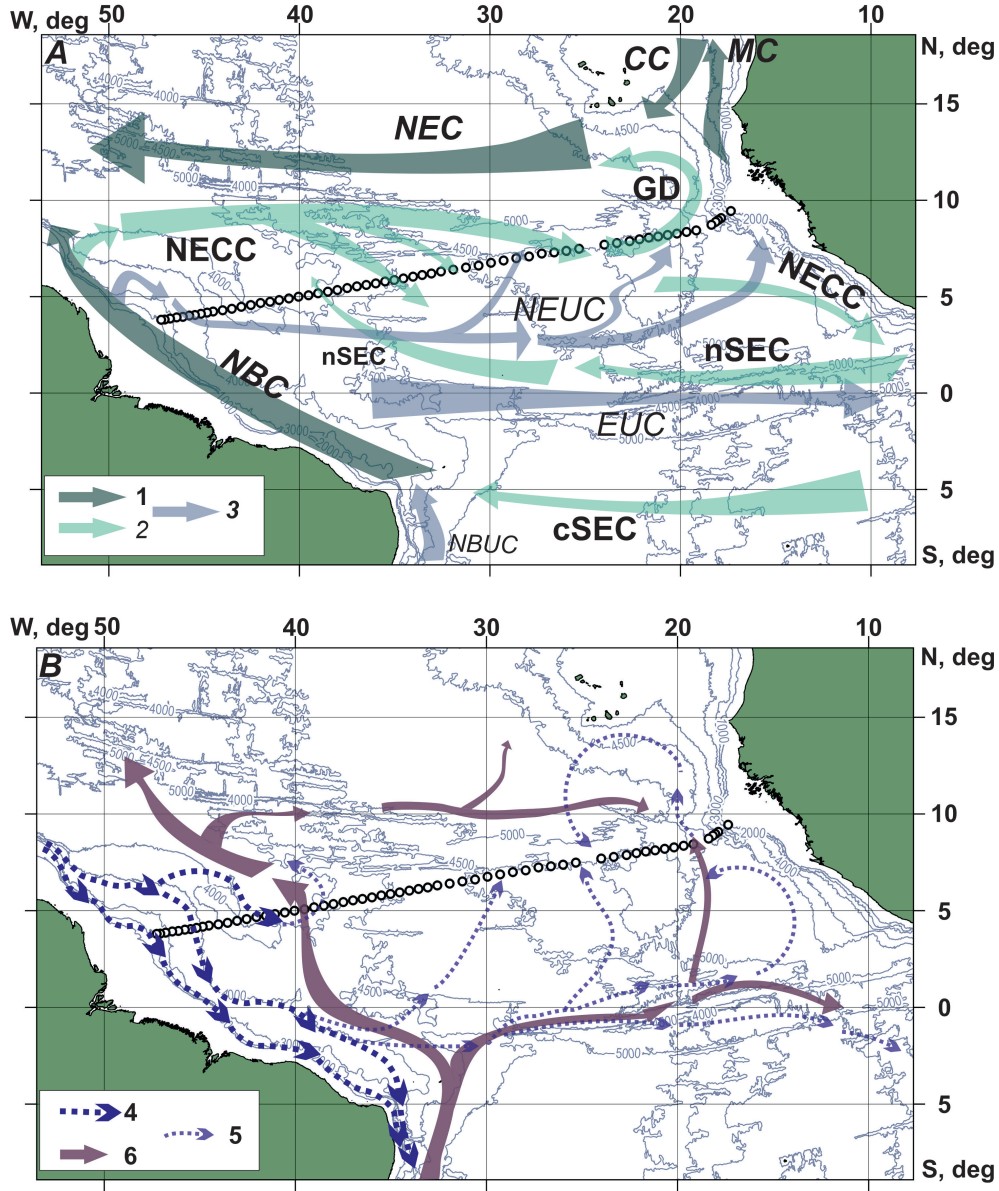

**Figure 2.** Generalized circulation schematics within the study area: surface and subsurface layers (A) and deep and bottom layers (B), where 1 — surface currents (from 0 up to 100 m), 2 — subsurface currents (from 100 up to 500 m), 3 — joint surface and subsurface currents (from 0 up to 500 m), 4 — the Deep Western Boundary Current (DWBC) cores, 5 — schematic Deep Western Boundary Current recirculation, 6— Antarctic Bottom Water. The references and current abbreviations are explained in the text. White circles — Ioffe-2000 transect stations.

Caused by the Atlantic trade wind belt, the North Equatorial Current is found in the North Atlantic from about 7° N to about 20° N (Schott et al., 2002) and is a broad westward flowing current that forms the southern limb of the North Atlantic subtropical gyre (Bourlès et al., 1999). The Guinea Dome (GD) is a permanent thermal upwelling dome (Rossignol and Meyrueis, 1964) centered at 9° N, 25° Wand at 10.5° N, 22° W in boreal summer and winter respectively (Siedler et al., 1992). The main existence conditions for the Guinea Dome are a cyclonic circulation composed of the eastward North Equatorial Countercurrent and North Equatorial Undercurrent along with the westward North Equatorial Current (Stramma and Schott, 1999).

According to Mittelstaedt (1991), reaching the African coast, the North Equatorial Countercurrent bifurcates and a part of its flow goes northwards, forming the Mauritania Current (MC). The summer — early autumn relaxation of the Northeast Trade winds (Lázaro et al., 2005) along with the North Equatorial Countercurrent strengthening are responsible for fact that the MC roughly reaches the 20° N parallel, just south of Cap Blanc. This process is associated with the cessation of upwelling and the Canary Current (CC) in this place (Mittelstaedt, 1991; Stramma et al., 2008).

### 3.2 Mid-depth ocean

Intermediate waters, namely, Antarctic Intermediate Water (AAIW) and Upper Circumpolar Water (UCPW) cross the equator northward mainly in the Western Atlantic between 500 m and 1200 m depth (Oudot et al., 1999). The vertical minimum of potential temperature (Reid, 1997) and more so the intermediate maximum of silicate concentrations (Oudot et al., 1999) are distinguishing features of Upper Circumpolar Water in the equatorial zone. Antarctic Intermediate Water is carried by several interchanging eastward and westward flowing currents in the equator area (Stramma and Schott, 1999; Brandt et al., 2006). Extended horizontal oxygen minimum zones exist in the eastern tropical Atlantic between 200 and 800 m due to microbial respiration of the large amounts of organic matter from the nearby North African upwelling area and weak water ventilation (Karstensen et al., 2008). The core of the North Atlantic oxygen minimum zone include South Atlantic Central Water and Antarctic Intermediate Water layers (Stramma et al., 2005). The major pathways to the northern oxygen minimum zones are subsurface eastward jet currents located south of 10° N: the Northern Intermediate Counter Current at 2° N, the North Equatorial Undercurrent at 4° N, and the northern branch of the North Equatorial Countercurrent at 8–9° N (Stramma et al., 2005; Karstensen et al., 2008; Brandt et al., 2010), while the core of the northern oxygen minimum zone lies further north. The link between these subsurface jet currents and the oxygen minimum zones is formed simultaneously by the cyclonic circulation around the Guinea Dome, a general uplift of the isopycnals within the surface and subsurface levels (Siedler et al., 1992; Lázaro et al., 2005).

### 3.3 Deep ocean

North Atlantic Deep Water (NADW) is formed in the North Atlantic by convection and mixing and transported southward. North Atlantic Deep Water penetrates into the Equatorial Atlantic at depths almost similar to the Deep Western Boundary Current, and bifurcates into an eastward and southward flow, correspondently into the open ocean and along the western boundary (McCartney, 1993). According to Wüst (1935), upper North Atlantic Deep Water (UNADW) can be recognized by

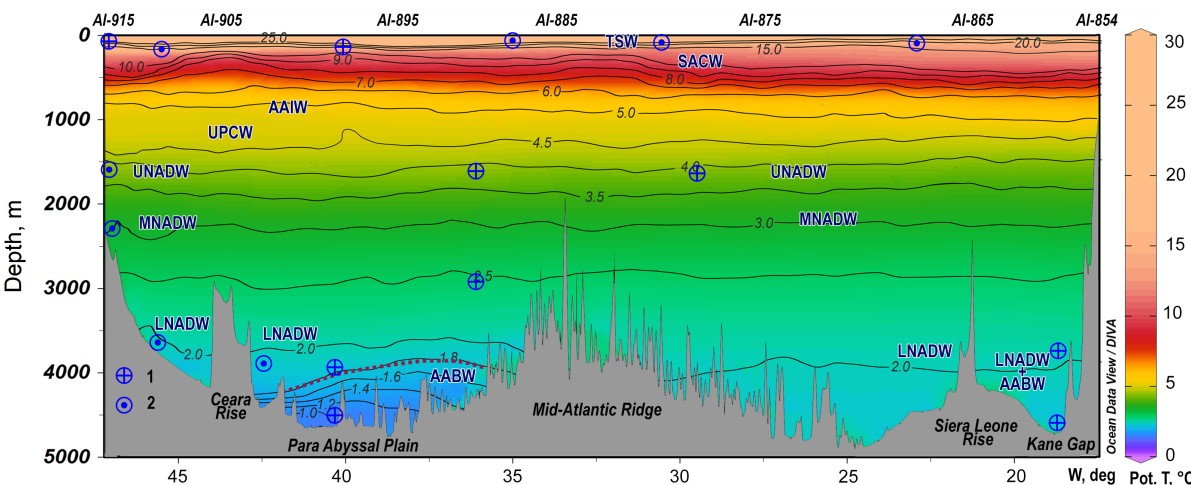

**Figure 3.** Distribution of the potential temperature (°C ) on the Ioffe-2000 transect (after Sokov et al. (2002)), done with Ocean Data View (Schlitzer, 2018). Numbers indicated in panel bottom left. 1 — northward current, 2 — southward current. For water mass abbreviations see text. Bottom topography from Sokov et al. (2002). The interpolation was done via the DIVA gridding (Barth et al., 2010) with 25 permille X-scale length and 25 permille Y-scale length. Dashed magenta line indicates$^{\sigma}4 > 45.90$ kg m$^{-3}$.

the mid-depth salinity maximum (1500–2000 m deep), while middle North Atlantic Deep Water (MNADW) at 2000–2500 m
deep and 3700 m deep lower North Atlantic Deep Water (LNADW) are recognized by the oxygen maxima. In Molinari et al.
(1992), waters that had been recently ventilated are described as having had contact with the atmosphere either directly or
indirectly by mixing on the time scale of the measurable transient tracer — chlorofluorocarbon. The highest values of one of
these chlorofluorocarbonss (F11 — trichlorofluoromethane) are found in lower North Atlantic Deep Water (Rhein et al., 1998).

Two cores of recently ventilated (with higher levels of F11) North Atlantic Deep Water from the northern hemisphere are
advected along the boundary and east to the Mid-Atlantic Ridge Molinari et al. (1992). A Deep Western Boundary Current
shallow core is centred at about 1500 m and a deeper one — at about 3500 m. The upper core of high F11 content, limited
by the 3.2 and 4.7°C potential temperature isotherms, is typically associated with the deeper core, which is limited by the 1.8
and 2.4°C isotherms. The Ceara Rise blocks the flow of the Deep Western Boundary Current waters with potential temperature
below 1.8°C to the equator, causing it to recirculate back to the north (Fig. 3).

Denser Antarctic Bottom Water (AABW) flows northwards through Atlantic Ocean basins and generally lies beneath North
Atlantic Deep Water at the bottom. The velocity field of Antarctic Bottom Water (water with $^{\sigma}4 > 45.90$ kg m$^{-3}$) is influenced
by the overlying Deep Western Boundary Current (Rhein et al., 1998) (see Fig. 3). The Antarctic Bottom Water is older
than North Atlantic Deep Water and has low F11 concentrations (Touratier et al., 2005) or even negligible in the Western
Atlantic (Reid, 1994). According to Rhein et al. (1998), Antarctic Bottom Water bifurcates at the equator: roughly 30 percent
of the northward Antarctic Bottom Water flow comes to the Guiana Basin through the Equatorial Channel at 35° W. A large
proportion of Antarctic Bottom Water turns east through the Romanche Fracture Zone to the Eastern Atlantic, with a possible

recirculation in the Brazil Basin. The sloping topography of Guiana Basin as well as the limitation of the Antarctic Bottom Water upper boundary by the strong eastward flowing North Atlantic Deep Water cause the northward directed geostrophic flow of Antarctic Bottom Water to be most pronounced only to the east of the Ceara Rise. Passing Ceara Rise, Antarctic

Bottom Water flows to the central part of Guiana Basin. The high level of F11 (> 0.07 pmol kg$^{-1}$) from lower North Atlantic Deep Water affects the part of Antarctic Bottom Water closest to the Ceara Rise, thus this Antarctic Bottom Water branch also experiences a rise in F11 content in comparison with its far eastern branch. Similar conditions are found to the north of Ceara Rise (Rhein et al., 1998). According to Whitehead Jr and Worthington (1982) and Rhein et al. (1998), the core of Antarctic Bottom Water is located to the east of Ceara Rise at 43.3° W.

Antarctic Bottom Water mixes with lower North Atlantic Deep Water in the fracture zones at equator, so Antarctic Bottom Water in the Eastern Atlantic differs from the similar water in the Western Atlantic (Rhein et al., 1998). There are two important features for the Eastern Equatorial Atlantic (McCartney et al., 1991). First, the potential flow of Antarctic Bottom Water through the Kane Gap from the Sierra Leone Basin to the Gambia Abyssal Plain and back and second, the potential Antarctic Bottom Water flow through the Vema Fracture Zone.

According to Arhan et al. (1998) the eastward transport of North Atlantic Deep Water along the equator is weak, but nevertheless plays a leading role for the maintenance of the equatorial tongue of upper North Atlantic Deep Water. Even though the eastward mean velocities are weak in the equatorial tongues of North Atlantic Deep Water, the question is of the ultimate fate of this flow as it reaches the African continental slope. There are two cyclonic flows in the deep layers of Eastern Atlantic, associated with North Atlantic Deep Water — the Gambia Basin one and the Sierra Leone Basin.

## 4  Results and discussion

### 4.1  SPM concentration

The distribution of SPM within the Ioffe-2000 transect is presented in Figure 4, while background hydrophysical and hydro-chemical conditions were described in detail in Sarafanov et al. (2007). The measured volume SPM concentrations varied from 0.01 to 0.40 ppm within the Ioffe-2000 transect and up to 4.1 ppm at the stations at the Northwest African upwelling area. The

work of McCave (1983) contains both the partly comparable dataset of SPM volume concentrations with a slightly wider size range (1.26–32 $\mu$m), and the apparent particle densities, that lies between 1.65 and 2.23 mg mm$^{-3}$. Applying these apparent densities to our volume SPM concentrations, we got the implied weight concentrations about 0.016 to 0.35 mg L$^{-1}$ (up to 0.8 mg L$^{-1}$ in exceptional circumstances), which agrees with Brewer et al. (1976) and Gardner et al. (1985a).

### 4.1.1  Upper ocean

In the upper layers, most commonly, the vertical distribution of SPM has a surface maximum, decreasing exponentially towards the deep. The main reason for this is the fact that the upper ocean contains both the external SPM sources (river discharge, atmospheric input) and the internal SPM sources (primary production) (Chester, 1990). Another reason for particle accumu-

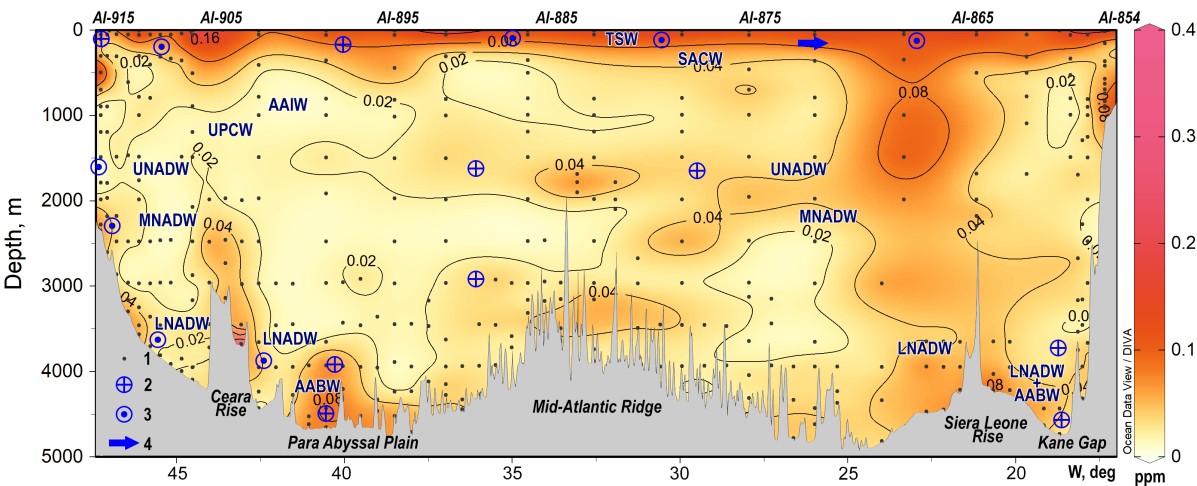

**Figure 4.** Distribution of the SPM volume concentration (ppm) on the Ioffe-2000 transect, done with Ocean Data View (Schlitzer, 2018). Numbers indicated in panel bottom left: 1 — sampling points, 2 — northward current, 3 — southward current, 4 — eastward current. Isolines are in geometric progression, for water mass abbreviations see text. Bottom topography from Sokov et al. (2002). The interpolation was done via the DIVA gridding (Barth et al., 2010) with 25 permille X-scale length and 65 permille Y-scale length.

lation in the surface layer may be the pycnocline, which slows the sedimentation (Emelyanov, 2005). Accumulation in the surface layer may be the existence of the pycnocline at the bottom of the surface layer, which slows the sedimentation. The
"sedimentation barrier" can be seen as a visible pycnocline at depths of 50–150 m to the west of the 30° W and 0–100 m to the east of this point on the Ioffe-2000 transect (Sarafanov et al., 2007). It was shown for the Eastern Equatorial Atlantic that the largest particulate elemental and mass concentration gradient occurs at the base of the mixed layer, where the particle and organism maximums are located. The source of the vast majority of particles in the upper open ocean waters is internal and consists basically of particulate organic matter (POM), $CaCO_3$, and biogenic silica. A large proportion of the POM produced
in surface waters is regenerated in the euphotic zone. Although the net transport of organic matter has to be downward to fuel the lower layers of the water column, there is also an upward component to transport. Positively buoyant particles, including lipid-rich eggs, larvae and, possibly, carcasses of deep-sea animals are examples of particles which undergo upward transport (Smith et al., 1989).

The Ioffe-2000 transect did not reach the SPM-rich shelf waters, yet the coastal SPM source has caused a local SPM
maximum at the margins of the transect at a depth of 300–900 m (up to 0.18 ppm in the west and up to 0.09 ppm in the east). The eastern edge of the Ioffe-2000 transect is located above the gentle slope between 200 m and 2000 m named Guinea Marginal Plateau (Egloff, 1972), adjacent to the high productive Guinea shelf (Vladimirov et al., 1990; Burlakova et al., 1997). Strong currents (Mittelstaedt, 1991; Stramma et al., 2005, 2008) and implied internal wave activity, based on significant density gradients on the shelf (0–200 m) (Sarafanov et al., 2007), may provide a framework for the SPM lateral transport from the shelf
along the gentle Plateau slope. It was the Amazon River that caused the more pronounced rise in SPM concentration at the

western edge of transect. It is well known that the surface SPM transport from the Amazon River to the open ocean turns to the northeast alongside the coast (Gibbs, 1974). According to Sarafanov et al. (2007), the warm, high-saline and silicate-poor upper waters observed on the Ioffe-2000 transect above 400–500 m were advected mostly by the North Brazil Current. The North Brazil Current core was located immediately westward of the studied transect near the shelf break (Johns et al., 1998).

A signature of the North Brazil Current eastern periphery is located between 46° W and 47° W at depths of 150–500 m (Sarafanov et al., 2007). The SMP concentrations within the North Brazil Current are relatively low (stations AI-910–AI-912). The main evidence of the SPM supply from the Amazon River to the shelf and the continental slope is a huge sediment body — the Amazon Cone. Bottom SPM maxima of Amazon River origin are able to form INL falling from the shelf break. INLs are mostly caused by the detachment of BNLs over the breaks in the slope (McCave et al., 2001). One of the INLs was apparently

observed during the Ioffe-2000 transect as a significant rise in the SPM concentration below the North Brazil Current.

Sarafanov et al. (2007) shows the North Equatorial Undercurrent equatorward pathway at 44–46° W. We have suggested that such V-shaped isopycnals at 46° W between the North Brazil Current and North Equatorial Undercurrent (Sarafanov et al., 2007, Fig. 3A) correspond with the aforementioned large eddy-like feature, the semi-permanent "Amazon Eddy" (Bruce et al., 1985).

The northern branch of the South Equatorial Current (nSEC) is associated with the northwestward or northward near-surface flow at 38–41° W, which recirculates northward into the North Equatorial Countercurrent. Probably, the local surface maximum of the SPM concentration at 44° W was caused by the northern branch of the South Equatorial Current and North Equatorial Countercurrent convergence. The surface SPM maximum was also noted in the most intense cross-transectal flow of the North Equatorial Countercurrent at 35–37° W (Sarafanov et al., 2007).

The eastward rise of the isopycnals at 28–25° W and their deepening at 24.5–21.5° W at 150–500 m indicates the southern periphery of the Guinea Dome on the Ioffe-2000 transect (Sarafanov et al., 2007, fig.3A). Immediately east of the Guinea Dome (21.5–18° W) the S-shaped double rotation (a combination of the cyclonic and anticyclonic structures) of the North Equatorial Undercurrent (Stramma et al., 2008, fig.12) occurred. It is known that the Guinea Dome influences biological activity, so the chlorophyll-a concentration is high in this region and linked to the intensity of the doming (Signorini et al., 1999; Pradhan

et al., 2006; Doi et al., 2009). The local SPM maximum below the Guinea Dome area within the Ioffe-2000 transect agrees with this data. Overall, the SPM distribution within the Ioffe-2000 transect's upper layer mainly reflects surface circulation patterns (Biscaye and Eittreim, 1977) as well as particle transport (vertical and horizontal).

### 4.1.2 Mid-depth ocean

According to toSarafanov et al. (2007), the intermediate waters on the Ioffe-2000 transect were recorded as an area of low in
salinity and rich in silicate waters, which extends at depths of 400–1200 m. The Antarctic Intermediate Water main entrance to the Northern Hemisphere is located in the western part of the transect (west of 36° W). Both Antarctic Intermediate Water and Upper Circumpolar Water do not come along the Brazilian slope while flowing from the western equatorial basin to the tropical North Atlantic, but rather flow in the interior ocean.

Usually the intermediate waters correspond with the upper part of the clear water minimum. Clear water is referred to the minimum in the SPM concentration that is commonly located above the BNL (Eittreim and Ewing, 1974; Biscaye and Eittreim, 1977). Concentrations of the SPM within this layer are 10–100 times lower than in the upper ocean but still reflect, in general, a similar geographic distribution with high values in the coastal zone and the lowest values at the open ocean. As was mentioned above, the clear water layer SPM concentration reflect, also, the lateral supply of the SPM from the coastal areas (Biscaye and Eittreim, 1977). However, the intermediate waters on the Ioffe-2000 transect were not influenced by the lateral source too strongly, so the SPM concentrations were low (0.01–0.03 ppm).

On the contrary, the wide area below the Guinea Dome (19–25° W) is noteworthy due to extremely high SPM concentration for this clear water layer, which primarily reflects the pattern of productivity in the upper waters (Eittreim et al., 1976). We suggest that this SPM-rich area was caused by the extra high bioproductivity levels within the Northwest African upwelling (Fig. 5) and, probably, the Guinea Dome area. The recent studies of the Guinea Dome (?) surface layer showed high SPM concentrations (from 0.1 and even up to 0.7 mg L$^{-1}$) for the low season of the Guinea Dome development. Equatorward transport of the POM is carried out by the interaction of the CC and the aforementioned system of equatorial currents. Additional explanation for the POM supply is the low dissolved oxygen level (oxygen minimum zones signature), that was observed at 300 to 700 m depth about 20–25° W and further westward to the Mid-Atlantic Ridge (Sokov et al., 2002). Not only is the oxygen minimum caused by the POM supply, but it also prevents organic particles from destruction.

### 4.1.3 Deep ocean

It is known that the vertical and zonal structure of the deep and bottom waters, i.e. North Atlantic Deep Water and Antarctic Bottom Water, in the Eastern Atlantic are significantly more homogeneous than in the Western Atlantic. The Eastern Atlantic generally shows low turbidity (SPM concentration) levels within the North Atlantic Deep Water and Antarctic Bottom Water in comparison with the Western Atlantic (Eittreim et al., 1976). The AABW transport in the deep layers of Eastern Atlantic is almost negligible in comparison with the intense northward flow of Antarctic Bottom Water to the east of the Ceara Rise in the western basin, that was noted during the Ioffe-2000 transect (Sarafanov et al., 2007).

*The Eastern Atlantic.* The research to the east of Mid-Atlantic Ridge at the Ioffe-2000 transect allowed us to note two peculiarities. The first one was the northward North Atlantic Deep Water flow through the Kane Gap into the Gambia Basin, while the second one pointed out the dominance of lower North Atlantic Deep Water in the near-bottom layer west of the Sierra Leone Rise.

The deep waters correspond with the lower part of the clear water minimum. However, abnormally high SPM concentrations for these depths were noted above the Sierra Leone Rise (Fig. 4). This SPM concentration anomaly was connected with another above it in the mid-depth ocean. There is also a slight horizontal "tail" with increased SPM concentration spreading almost to the Mid-Atlantic Ridge. The probable reason for existence of the "tail" is the cyclonic flow over the Gambia Abyssal Plain (Arhan et al., 1998). We explain this "vertical anomaly" by using the ballast hypothesis (Armstrong et al., 2001; Louis et al., 2017), which is based on the correlation between the fluxes of particulate organic carbon (POC) and minerals in the deep ocean. According to hypothesis description in Van der Jagt et al. (2018), biogenic aggregates are ballasted with biogenic and

lithogenic minerals, so their sinking velocities increase. Minerals start to get attached to aggregates both in the surface layer and deeper: during the aggregates formation or when aggregates "scavenge" mineral particles while sinking. The fact that ballasted

aggregates sink with higher speed is the main reason for them to be remineralized at a greater depth in comparison with non-ballasted aggregates. Different aggregates are incorporated with different number of minerals due to different stickiness, which is controlled by the amount of transparent exopolymer particles (Alldredge et al., 1993). Precursors of transparent exopolymer particles are produced mainly by phytoplankton (Passow and Alldredge, 1994). The amount of transparent exopolymer particles in the ocean is huge, they occur as "sticky" gels (Passow, 2002), so transparent exopolymer particles act as a glue matrix for

other solid particles (i.e., detritus), forming larger aggregates ("marine snow") and playing a crucial role in the carbon export from the surface to the deep ocean (Passow et al., 2001).

The SPM in regions with high external supply (for example, the North Atlantic) with their high sedimentary dust inputs consists mostly of lithogenic material. For instance, the surface waters off Cap Blanc (Iversen and Ploug, 2010) and Guinea Dome (Bubnova et al., 2020) are exposed to the Saharan dust. The Saharan dust deposition may cause the abiotic formation of

transparent exopolymer particles, which results in the aggregates formation and enhancing the POC export (Louis et al., 2017). Dust is able to ballast into marine snow aggregates (and fecal pellets) and it also increases the oceanic primary production due to input of dust-related micronutrients (Van der Jagt et al., 2018). Potential strength of the so-called "lithogenic carbon pump" (Bressac et al., 2014) potentially depends on the physical and chemical precursors of transparent exopolymer particles featured, which, in turn, stand on the composition and physiological state of the phytoplankton community. The Northwest

Africa offshore zone shows high levels of primary production and SPM concentrations (Fig. 5). This area is also exposed to abundant Saharan dust input. The resulting large number of ballasted aggregates is then transported equatorward via the CC and consistent flow of the North Equatorial Current, North Equatorial Countercurrent and Guinea Dome (equatorial currents system), reaching the Ioffe-2000 transect area.

The Van der Jagt et al. (2018) study showed that the total volume of aggregates increased ten times due to dust deposition,

while size-specific sinking velocities of the dust-ballasted aggregates increased two times. It was also shown that the dust-ballasted aggregates carried only 50 percent of the volumetric POC amount in comparison with non-ballasted aggregates. Thus, the overall POC flux driven by dust deposition may rise up to ten times due to the abundance of aggregates. As we remember, fast sinking aggregates are remineralized deeper leading to an increase in the POC fluxes to the deep sea layers (Van der Jagt et al., 2018), which preserves the high SPM concentrations down to the deep ocean. Such fast sinking ballasted

aggregates, which were transported from the Northwest Africa upwelling area, caused the abnormally high SPM concentrations above the Sierra Leone Rise in the Ioffe-2000 transect.

*The Western Atlantic.* The vast area to the west of the Mid-Atlantic Ridge within the Ioffe-2000 transect show higher (in comparison with the Eastern Atlantic) values of SPM concentrations, caused by Deep Western Boundary Current. It is well-known that there are two reasons for the bottom currents to maintain high loads of SPM: potential resuspension and, far more

importantly, selective deposition. The SPM may be transported far within the BNLs, as was shown for clay minerals (Petschick et al., 1996). The most intense BNLs in the western Atlantic are found in the southwestern Argentine Basin and northern North American Basin; while the lowest bottom water turbidity is located in the equatorial regions (Eittreim et al., 1976; Biscaye and

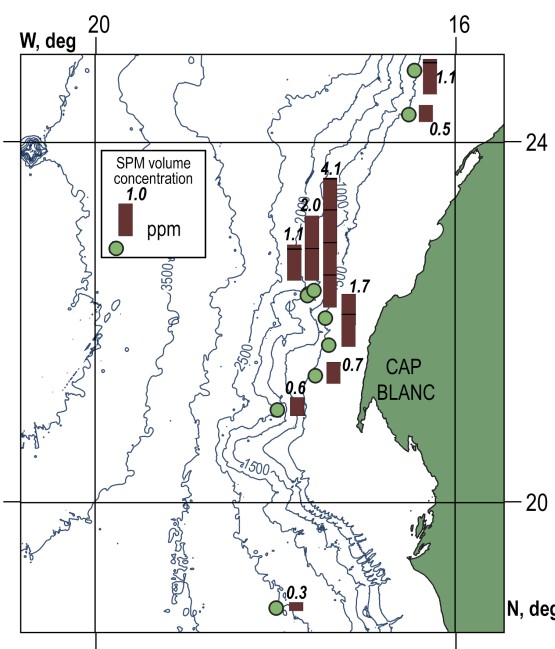

**Figure 5.** Schematic distribution of surface SPM concentrations (ppm) in the Northwest African upwelling area prior to the Ioffe-2000 transect. Isobaths every 500 m.

Eittreim, 1977). Nevertheless, bottom SPM concentrations west of the Mid-Atlantic Ridge are still higher in comparison with the eastern part of the Equatorial Atlantic. According to Sarafanov et al. (2007), the southward boundary flow of Upper North Atlantic Deep Water was traced in the western basin along the continental slope at the depths of 1400–1500 m. Middle North Atlantic Deep Water southward flow was noted at 1900–2600 m with the 100 km wide recirculation. The general lower North Atlantic Deep Water flow was noted at 3700 m depth. The deep cyclonic cell in the Guiana basin (McCartney, 1993; Arhan et al., 1998) at North Atlantic Deep Water depth is located near the Ceara Rise western slope and west of the Mid-Atlantic Ridge between 42° W and 36–37° W. This northward recirculation of lower North Atlantic Deep Water joins the northward flow of Antarctic Bottom Water.

The noticeable BNL in the western basin is located at 1900–2600 m above the American slope (the Amazon Cone). We suggest that the main reason for its occurrence is the interaction between the shallow core of the Deep Western Boundary Current and the Amazon River solid load.

The weak INL deepening from the slope to the east was noted at a depth of 3200–3700 m (Fig. 6). It is enhancing above the Ceara Rise as a result of the Deep Western Boundary Current intensification due to the interaction with bottom topography. This INL can be traced to the east from the Ceara Rise (4300 m depth and 42.5° W), where the deep Deep Western Boundary Current core is marked by high F11 level.

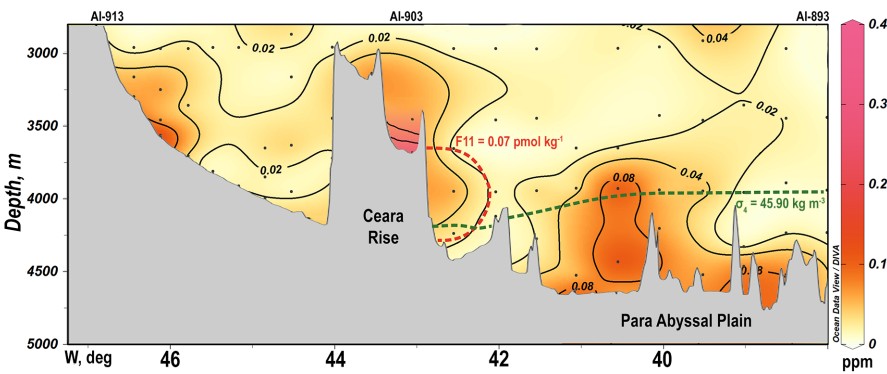

**Figure 6.** The SPM volume concentration at the western part of the Ioffe-2000 transect (magnified part with more detail of the Figure 4): green dashed line — the upper boundary of Antarctic Bottom Water; red dashed line — the Deep Western Boundary Current boundary, based on the F11 (after Rhein et al. (1998), 4.5° N transect). Isolines in geometric progression.

The main reason for the SPM concentration rise at the depth of 3500–4300 m between 41.5° W and 39° W with the peak at 40.5° W is the northward recirculation of SPM-rich lower North Atlantic Deep Water.

The general northward pathway of Antarctic Bottom Water ($\sigma 4 > 45.90$ kg m$^{-3}$) across the Ioffe-2000 transect is located in the Para Abyssal Plain bottom layer. Antarctic Bottom Water core was found in the 40–42° W troughs, flowing along the Ceara Rise eastern slope (Sarafanov et al., 2007). Studies (Whitehead Jr and Worthington, 1982; Rhein et al., 1998) show the Antarctic Bottom Water core to be separated from the bottom and located at a depth of 4400–4500 m. Antarctic Bottom Water has high SPM concentrations in comparison with the clear water layer, mostly in the 40–41° W troughs. The SPM concentration

maximum within Antarctic Bottom Water (Fig. 6) is aligned with the Antarctic Bottom Water core separated from the bottom (4400–4500 m). The reason for the separation may be the bottom rise ("dam"), which is located up-stream Antarctic Bottom Water flow (Whitehead Jr and Worthington, 1982). The "dam" blocks the lower part of Antarctic Bottom Water flow and coinciding BNL at the depth of 4500 m, creating the INL from BNL. One of the most notable similar nephelometric features was described for the Puerto Rico Trench near the Navidad sill (Tucholke and Eittreim, 1974). The BNL to the east of the

Antarctic Bottom Water flow core is likely caused by Antarctic Bottom Water southward recirculation (Sarafanov et al., 2007).

     The extremely high SPM concentration (0.36 ppm) was noted in bottom layer above the Ceara Rise (AI-903, depth 3678 m, 18 meters above bottom). The potential reason for that may be interaction between the SPM-rich Deep Western Boundary Current and a local feature in the bottom topography. Figure 7 shows, that AI-903 station was conducted near the terrace break (3800 m). The terrace slope is low-angle (0.7°), while below the terrace break lies a far steeper (3.1°) slope (3800–4300 m).

This 15-km wide area with the steep incline perturbs the homogeneity of the Deep Western Boundary Current core: the part of the current above the flat terrace slows down, while the steeper slope causes a higher bottom current speed (Bowden, 1960). The current slow-down above the terrace is able to cause an eddy. The V-shaped terrace (see 3700 m isobath) allows the assumption that this eddy is the topographically trapped one. The eddy originating from the SPM-rich Deep Western Boundary Current

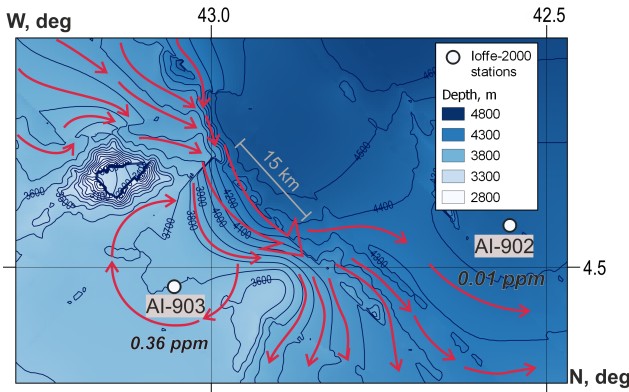

**Figure 7.** The Ceara Rise SPM maximum: bottom topography (GEBCO Bathymetric Compilation Group, 2020) and the Deep Western Boundary Current (red lines).

causes an additional SPM concentration boost near the bottom. The proposed hypothesis undoubtedly needs to be confirmed by additional research.

*The Mid-Atlantic Ridge area.* High levels of the SPM concentration were also noted above the Mid-Atlantic Ridge (Fig. 4). There is the northward recirculation of North Atlantic Deep Water rich in the SPM (Sarafanov et al., 2007), which was noted both sides of the Mid-Atlantic Ridge at 36–40° W and 29–32° W. Another type of currents in the area is tidal currents (Morozov, 2018). According to Lavelle (2012), both these currents may be accelerated along the flanks of the ridge and represent a significant stirring mechanism for abyssal flow and cause the SPM concentration rise. Moreover, the axial region of the Mid-Atlantic Ridge including the rift zone experiences high seismicity and bears a large number of earthquake epicentres as well as the sulfide mineralization zones of various origins and bedrock zones showing strong hydrothermal imprints (Mazarovich and Sokolov, 2002); in particular in the Sierra Leone fracture zone, i.e. immediately below the Ioffe-2000 transect (see Fig. 1). Both high seismicity and strong hydrothermal activity may also be able to cause increased SPM concentrations in the bottom layer.

## 4.2 Grain-size of the SPM

The size distribution of suspended particles is generally considered to be a function of the source and nature of the particles, processes of aggregation and the "age" of the suspension (McCave, 1983, 1985). SPM grain sizes in cumulative number distribution from Ioffe-2000 transect follow a hyperbolic Junge distribution with slopes between –1.9 and –3.7 (Fig. 8), which correspond with McCave (1975), where slopes were mainly between –2.4 and –3.6. The difference occurs due to the fact that McCave (1975) had a wider size spectrum. The overall cumulative number distribution for Ioffe-2000 transect were recorded as a wavy line with an overall slope approximating –2.7 which is similar to McCave (1984, 1986); Nowell et al. (1985).

One of the first detail analysis of the SPM volume-size distributions was conducted during the HEBBLE project in abyssal waters (the Nova Scotia Rise, 1977–1984) (McCave, 1983; Nowell et al., 1985; McCave, 1985). The sizes measured by a

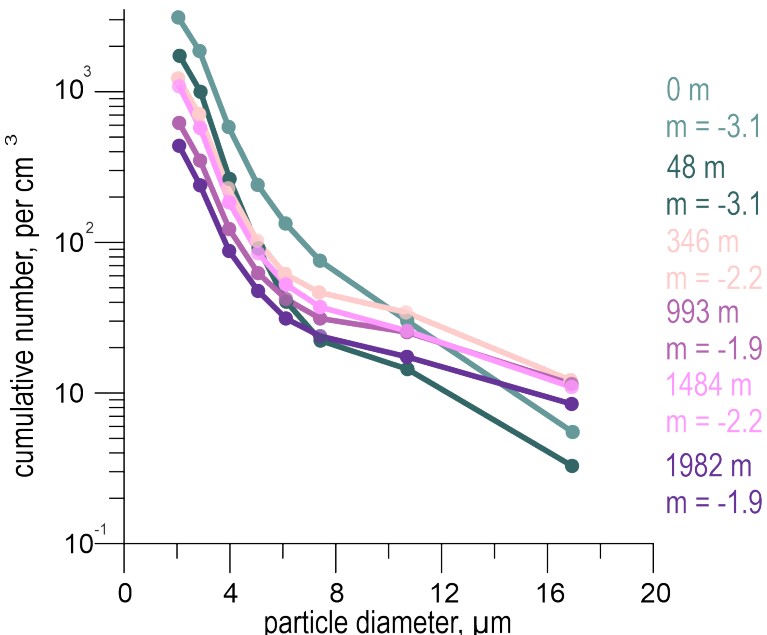

**Figure 8.** Cumulative particle number distribution for AI-868 station showing both "old" suspension in the upper ocean with the slope of the distributions close to –3 and "fresh" in the water column with steep slopes about –2. Station AI-868 is a part of the SPM-rich "vertical anomaly". X-axis is particle diameter, Y-axis is cumulative particle number in log10 scale.

360 Coulter Counter in this project mainly showed the bimodal structure with a fine mode between 2 and 10 μm and a coarse mode between 20 and 60 μm.

Since the Coulter Counter aperture-tube truncation during the Ioffe-2000 was 70 μm, it was impossible to count particles up to 32 μm, so only the "left" part of the coarse mode is visible on Ioffe-2000 data (Fig. 9).The volume-size distributions display: a) an elevated proportion of the 7–21 μm fraction, suggesting the coarse mode, b) a fine-mode with the peak size ranges about 365 2–4 μm, and c) an intermediate mode at 8–13 μm, that was not mentioned in the HEBBLE project.

The most representative bimodal volume-size distributions in our study are shown in Figure 9. According to McCave (1983, 1984), there are two most effective physical mechanisms for changing the size distributions: aggregation of small particles (below 1.5–8 μm) by Brownian motion and collection of smaller particles by turbulent agitation with larger particles. Thus, the fine mode mainly consists of aggregates. The more turbid suspensions according to Gardner et al. (1985b) contain a 370 coarser fine mode (~ 8 μm) than the lower concentration suspensions with a mode of 2–3 μm. According to McCave (1984), oceanic particle size volume distribution shows pronounced peaks in BNLs with active resuspension and in surface waters with active primary production. The aggregation rate of 1.5–8 μm particles in the upper ocean depends on filtration by zooplankton. The fine mode in our data occurs within 3–8μm and in mostly in samples with low concentration of about 0.01 ppm (188 samples in total).

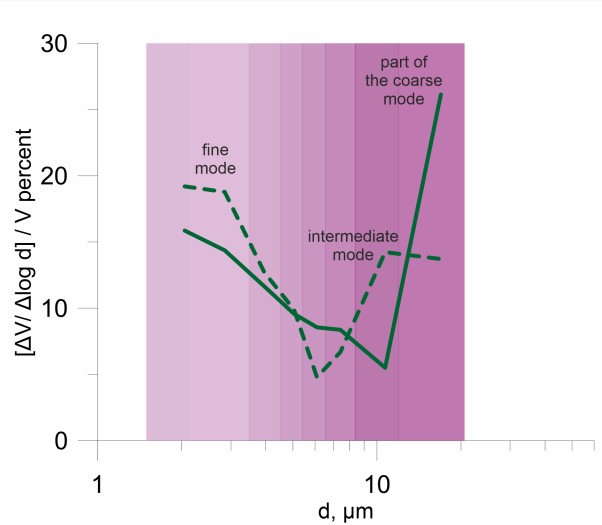

**Figure 9.** The typical examples of the bimodal volume-size SPM distribution. X-axis is particle diameter, Y-axis is probability density function. Green solid line — dominance of aggregates (station AI-903, 3678 m); dashed green line — intermediate mode, caused mostly by phytoplankton, detritus and transparent exopolymer particles (station AI-887, 3717 m). Size intervals are marked with color.

The upper ocean is the area, where the particle production and modification of the initial size distributions occurs controlled by turbulence. Increased shear in pycnoclines and internal wave-driven turbulence would promote aggregation and lead to a further smoothing of the size distribution (equivalent to a cumulative particle number distribution with a slope of –3) of SPM (McCave, 1984). The horizontal supply of SPM to the mid-water in the open ocean is relatively slow, therefore, a part of fine material in SPM is old (years old), so the size distribution should not have significant peaks. The flat size distributions are best explained by the sub-equal production of particles of different sizes. The coarse mode (20–60 μm), according to McCave (1985), is a result of aggregation, where a significant role is played by mucus. Lately, as was mentioned above in the paper, mucus has started to be referred as transparent exopolymer particles. There are no strong vertical gradients of mucus concentration suggesting a bottom source, so it is presumed that the material results from surface productivity and has settled rapidly. This viewpoint is supported in Emery and Honjo (1979), where the same type of material from surface waters is described and called an "(organic) film". It was shown that the film is most abundant in upwelling areas. The "left" part of the coarse mode was noticed within the "vertical anomaly", as well as in the BNLs (88 samples in total). The intermediate mode (8–13 μm) was dominant in surface and intermediate waters (50–3000 m depth) in the samples with high SPM levels (0.3 ppm) (25 samples in total). Its occurrence is supposedly caused by high concentration of phytoplankton with cell size less than 10 μm, as well as newborn transparent exopolymer particles and detritus. The vertical distribution of different fraction sizes of phytoplankton in the euphotic zone of the Equatorial Atlantic shows that the dominant plankton cell size at the 50 m depth is 3–20 μm (11.5 μm on average) (Herbland et al., 1985). This intermediate mode could not be revealed during the HEBBLE project, because the project was aimed exclusively at abyssal waters.

# 5 Conclusions

The Ioffe-2000 transect presented a "screenshot" of volume SPM distribution in the northern part of the Equatorial Atlantic
against the background of hydrographic data. Alongside the general agreement with the three-layer model of the ocean SPM
distribution, there were some large regional non-uniformities of SPM concentrations and grain size. The international studies
of last two decades allowed us to explain those non-uniformities:

First. The most noticeable anomaly is a wide area of high SPM concentrations above the Sierra Leone Rise spreaded from
the upper ocean, through the intermediate layer of clear water and down to the ocean bottom. We suggest that this "vertical
anomaly" originated from the Northwest African upwelling area, where the aggregation process exists, since the plankton
exposed to abundant Saharan dust form a large number of ballasted aggregates. The aggregates were lately transported equa-
torward via the Canary Current and equatorial currents system (including the Guinea Dome). The vertical SPM concentration
anomaly, revealed in our study, agrees with the fact that higher aggregate numbers and higher sinking velocities may increase
the sediment transport to the bottom and since the biogenic aggregates are rich in carbon, the carbon export to deep ocean
layers will also increase.

Second. Deep and bottom waters of the western part of the Equatorial Atlantic show increased levels of the SPM con-
centrations. The INL deepening from the American slope to the east was noted at depths of 3200–3700 m. The high SPM
concentrations in the bottom layer above the Ceara Rise is caused by the interaction between the SPM-rich Deep Western
Boundary Current and a local feature of bottom topography. Antarctic Bottom Water has increased SPM concentrations mostly
in the 40–41° W troughs. The SPM maximum is located in the Antarctic Bottom Water core (4400–4500 m), separated from
the bottom. The explanation for this may be that the bottom rise located up-stream serves as a "dam", blocking the lower part
of Antarctic Bottom Water flow and its BNL.

Third. The sizes of suspended particles in the cumulative number distribution follow a hyperbolic Junge distribution with
slopes between –1.9 and –3.7; the averaged slope was approximated as –2.7, which fits into the general understanding.

The volume-size distributions displayed two modes -comparable to those found in HEBBLE: a fine-mode 2–4 μm and an
increased proportion of the 7–21 μm fraction, suggesting the coarse mode (20–60 μm). There was an additional intermediate
mode 8–13 μm. It is well known, that the fine mode is mainly represented by aggregates formed from small particles by
Brownian motion. The weakly pronounced fine mode could be suggesting a relatively old age of SPM and low concentrations.
The coarse particles (7–21 μm) likely represent aggregates with biogenic mucus (transparent exopolymer particles), which
plays a significant role in particle aggregation. The additional intermediate mode (8–13 μm) is dominant in the SPM-rich
surface waters and is presumable the result of phytoplankton growth and the initial transparent exopolymer particles formation.

*Data availability.* The coordinates (Lat, Lon) of the Ioffe-2000 sampling stations, depths of samplings and volume suspended particulate
matter concentrations are given in Supplement 1.

*Author contributions.* Vadim Sivkov: conceptualization, formal analysis, investigation, resources, supervision, funding acquisition; Bubnova Ekaterina: methodology, data curation, visualization, formal analysis. All authors have read and agreed to the published version of the manuscript.

*Competing interests.* The authors declare no conflict of interest. The funders had no role in the design of the study; in the collection, analyses, or interpretation of data; in the writing of the manuscript, or in the decision to publish the results.

*Acknowledgements.* The preliminary data processing and surface ocean data interpretation were conducted with financial support of the state assignment of IO RAS (Theme 0128-2021-0016), the data interpretation and conclusions regarding the modern Antarctic Bottom Water circulation was supported by Russian Science Foundation (project 19-17-00246). The article was finished in kind memory of academician A.P. Lisitsyn (IO RAS) who helped with this research. Marina Ulyanova and Leyla Bashirova contributed with valuable advice and the fruitful discussions. Authors also would like to thank Professor Ian Nicholas McCave, who definitely made us to improve the paper significantly.

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
