# Peer review of "Distribution of suspended particulate matter at the equatorial transect in the Atlantic Ocean"

_Ocean Science, 2021_

## Author Response (AR1)

Dear Mr Hoppema!

We would like to thank both reviewers for their work as well as the editorial office. We have improved our manuscript in accordance with both comments. Corrections of English are marked in text in the track-changes file. We also have changed one of the grants, that supported this paper.

**Nick I. McCave comments**

14-21:

(1) In the introduction the authors mention the seminal work of Biscaye and Eittreim (1977) who ascribed thick bottom nepheloid layers (BNLs) to upward mixing of SPM, but they should also mention the lateral transfer arguments of Armi (1978) and McCave (1983), the 'separated mixed layer' model.

(2) We added both references

(3) The works of Armi (1978) and McCave (1983) pointed out the lateral advection of SPM, which occurs due to detaching of bottom mixed layer from the slope and leads to thickening and layering of BNLs.

52-56:

(1) With so few regional measurements of particle volume by Coulter counter one might ask whether the data presented here are 'correct', i.e. whether contamination has been avoided…

(2) After applying the apparent densities of suspended particles from the (McCave, 1983), we obtained the estimated weight SPM concentrations at the Ioffe-2000 transect. The according changes were put at the beginning of the Results Chapter.

(3) The work of McCave (1983) contains both the partly comparable dataset of SPM volume concentrations with a slightly wider size range (1.26—32 µm), and the apparent particle densities, that lies between 1.65 and 2.23 mg mm-3. Applying these apparent densities to our volume SPM concentrations, we got the implied weight concentrations about 0.016 to 0.35 mg L-1 (up to 0.8 mg L-1 in exceptional circumstances), which agrees with Brewer et al (1976) and Gardner et al. (2018).

174-184:

(1) The authors point to an influence of Amazon River sediments being more important than concentrations at the African end of the transect. Nevertheless, there is a marked high at the African end centred on about 800 m that the authors do not discuss and one wonders whether both might be due to internal wave activity on the upper slope.

(2) We added the lacking discussion

(3) The Ioffe-2000 transect did not reach the SPM-rich shelf waters, yet the coastal SPM source has caused a local SPM maximum at the margins of the transect at a depth of 300–900 m (up to 0.18 ppm in the west and up to 0.09 ppm in the east). It was the Amazon River that caused the

more pronounced rise in SPM concentration at the western edge of transect. It is well known that the surface SPM transport from the Amazon River to the open ocean turns to the northeast alongside the coast (Gibbs, 1974). The eastern edge of the Ioffe-2000 transect is located above the gentle slope between 200 m and 2000 m named Guinea Marginal Plateau (Egloff, 1972), adjacent to the high productive Guinea shelf (Vladimirov et al., 1990; Burlakova et al., 1997). Strong currents (Mittelstaedt, 1991; Stramma et al., 2005, 2008) and implied internal wave activity, based on significant density gradients on the shelf (0–200 m) (Sarafanov et al., 2007), may provide a framework for the SPM lateral transport from the shelf along the gentle Plateau slope.

225 et seq:

(1) The authors describe the high mid water concentrations extending down to the bottom over the Sierra Leone rise to the occurrence of aggregates ballasted with Aeolian dust. The concentration zone occurs 1000 km from the coast which is well beyond the zone of coastal upwelling-driven high productivity but does fall in the region of Sahara and Sahelian dust. The authors observe that this column of high concentration occurs under the Guinea dome, a permanent thermal upwelling dome with a cyclonic associated circulation. Other examples of high concentration columns are shown by Biscaye and Eitttreim over Bermuda rise and in the Argentine basin. It seems more likely that the authors observations are related to this circulation feature.

(2) According to Sarafanov et al. (2007), the Guinea Dome cyclonic circulation within the Ioffe-2000 transect was only noticeable within the upper 300 m of the water column, so we believe that it was incomparable with the DWBC that caused the high SPM concentrations shown by Biscaye and Eittreim over Bermuda rise and in the Argentine Basin. Naturally, the circulation in the area of our observations matters, yet we suppose that the entire system of currents, including the Guinea Dome, North Equatorial Current/Countercurrent/Undercurrent and the Canary Current plays a role mostly in the SPM transport from the highly productive area of the Northwest African coast and the region of Sahara and Sahelian dust.

(3) -

286-291:

(1) The intermediate nepheloid layer (INL) downstream of the 'dam' is similar to the INL demonstrated by Tucholke and Eittreim (1974) deep western boundary current flows over the Puerto Rico Trench.

(2) We did put the additional reference in our paper.

(3) One of the most notable similar nephelometric features was described for the Puerto Rico Trench near the Navidad sill (Tucholke and Eittreim, 1974).

301-306:

(1) Lavelle (2012) has shown the effect of midocean ridges on accelerating currents along their flanks, currents which are likely to then lead to resuspension and generation of nepheloid layers.

(2) We added Lavelle (2012) in our paper.

(3) High levels of the SPM concentration were also noted above the MAR. There is the northward recirculation of rich in the SPM NADW (Sarafanov et al., 2007), which was noted both sides of the MAR at 36–40° W and 29–32° W. Another type of currents in the area is tidal currents Morozov (2018). According to Lavelle (2012), both these currents may be accelerated along the flanks of the ridge and represent a significant stirring mechanism for abyssal flow and cause the SPM concentration rise. Moreover, the axial region of the MAR including the rift zone experiences high seismicity and bears a large number of earthquake epicenters as well as the sulfide mineralization zones of various origins and bedrock zones showing strong hydrothermal imprints (Mazarovich and Sokolov, 2002); in particular in the Sierra Leone fracture zone, i.e. immediately below the Ioffe-2000 transect (see Fig. 1). Both high seismicity and strong hydrothermal activity may also be able to cause increased SPM concentrations in the bottom layer.

310-346:

(1) It would be helpful to have a figure in which some of the size distributions were illustrated - cumulative number plots for example.

(2) We thought that the most typical volume-size SPM distributions will be sufficient, but we will add the cumulative number figure as well.

(3) Additional Figure 7.

[Figure]

Figure 7. Cumulative particle number distribution for AI-868 station showing both "old" suspension in the upper ocean with the slope of the distributions close to –3 and "fresh" in the water column with steep slopes about –2. Station AI-868 is a part of the SPM-rich "vertical anomaly". X-axis is particle diameter, Y-axis is cumulative particle number in log10 scale.

330-333:

(1) It is not clear how a large Brunt-Vaisala frequency in itself would lead to smoothing of a particle size distribution, but perhaps the authors wish to imply that there might be breaking internal wave-driven turbulence associated with frequency maxima that would promote aggregation.

(2) We paraphrased the paragraph in order to remove uncertainties.

(3) Increased shear in pycnoclines and internal wave-driven turbulence would promote aggregation and lead to a further smoothing of the size distribution (equivalent to a cumulative particle number distribution with a slope of -3) of SPM (McCave, 1984).

Additional references

Burlakova, Z. P., Eremeeva, L. V., & Morozova, A. L. (1997). Suspended matter in the estuaries of the Guinean shelf. physical Oceanography, 8(4), 269-283.

Stramma, L., Brandt, P., Schafstall, J., Schott, F., Fischer, J., & Körtzinger, A. (2008). Oxygen minimum zone in the North Atlantic south and east of the Cape Verde Islands. Journal of Geophysical Research: Oceans, 113(C4).

Vladimirov, V. L., Bezborodov, A. A., Martynov, O. V., Ovsyanyi, E. I., & Diallo, B. (1990). Relation between the depth of visibility of a white disk and the concentration of suspended matter in shelf waters off the Republic of Guinea. Soviet journal of physical oceanography, 1(6), 469-474.

**Second referee comment**

Fig. 2:

(1) A) It would be useful to add the approximate depth to the 1/2/3 notations, i.e. (numbers are just for illustration – please use the real ones): 1-surface currents (0-100m), 2-subsurface currents (100-300m), 3-deep surface currents (300-500m) A) and B): looks like isobaths are not at every 500m as stated in the caption. I see 1000, 2000, 3000, 4000, 4500, 5000. Please correct either capture or plots.

(2) We have changed both the caption of Figure 2 and Figure 2A itself.

(3)

[Figure]

Figure 2. Generalized circulation schematics within the study area: surface and subsurface layers (A) and intermediate, deep, and bottom layers (B), where 1 — surface currents (from 0 up to 100 m), 2 — subsurface currents (from 100 up to 500 m), 3 — joint surface and subsurface currents (from 0 up to 500 m), 4 — the Deep Western Boundary Current (DWBC) cores, 5 — schematic DWBC recirculation, 6— AABW. The references and current abbreviations are explained in the text. White circles — Ioffe-2000 transect stations.

Lines 95-100:

(1) The sentence "The main existence conditions for GD existence is a cyclonic circulation composed of the eastward NECC and NEUC along with the westward NEC (Stramma and Schott, 1999)." could use some editing to eliminate a repetition of word "existence".

(2) Corrected

(3) The main existence conditions for GD are a cyclonic circulation composed of the eastward NECC and NEUC along with the westward NEC (Stramma and Schott, 1999).

Fig. 3:

(1) Search radii? Show the mixed layer depth?

(2) We are afraid that showing the thin (100 m and less according to Sarafanov et al. (2007) upper mixed layer will not be illustrative, taking into account the fact that the overall depth in Figure 3 is 5000 m. Some information regarding gridding was added to the Figure 3 caption.

(3) The interpolation was done in the Ocean Data View software (Schlitzer, 2018) and the DIVA gridding (Barth et al., 2010) with 25 permille X-scale length and 65 permille Y-scale length.

Additional references

Barth, A., Alvera-Azcárate, A., Troupin, C., Ouberdous, M., & Beckers, J. M. (2010). A web interface for griding arbitrarily distributed in situ data based on Data-Interpolating Variational Analysis (DIVA). *Advances in Geosciences*, *28*, 29-37.

Lines 179-180

(1) Use station # for better location reference?

(2) We have corrected the paper.

(3) The SMP concentrations within the NBC are relatively low (stations AI-910–AI-912).

Line 197:

(1) "So the local SPM maximum in the GD area within the Ioffe-2000 transect correlate with this data" - show the correlation.

(2) The mathematical term "correlation" was a wrong choice for this sentence because there was no mathematical comparison.

(3) The local SPM maximum below the GD area within the Ioffe-2000 transect agrees with this data.

22.08.2021
[Figure]
 Ekaterina Bubnova

---

## Author Response (AR2)

Dear Mario Hoppema!
Thank you very much for your massive work with our paper!

To solve the problem of excessive number of abbreviations in the text we decided to use less of them since the table in our opinion seem to be more confusing for readers. See changes in the text. Yet we would like to remain SPM, BNL and INL abbreviations since they are well-known in sedimentology and it is convenient to search for them in the text. We also did mention abbreviation at figures in order to keep the pictures clearer.

The mentioned paper (Bubnova, E., Kapustina, M., Krechik, V., and Sivkov, V.: Suspended matter distribution in the surface layer of the East Equatorial Atlantic) has its English version, but the area of research for that paper was pretty small in comparison with the current paper (Fig.1), so the reference may be addressed only as one situational research of the Guinea Dome. We added addition reference in the Results – Mid-Ocean chapter.

[Figure]

**Fig. 1.** Research area. (*1*) Ship stations of cruise 33 of R/V *Akademik Nikolaj Strakhov*; (*2*) ship stations of cruise 8 of R/V *Akademik Ioffe*; (*3*) ship stations of cruise 32 of R/V *Akademik Ioffe*; (*4*) isobaths, m; (*5*) Equatorial Countercurrent; (*6*) Canary Current.

Figure 1. Research area of the reference paper.

(1) L2 Why not just introduce the abstract with the fact that you measured SPM on a transect in the year 2000?
(2) Corrected
(3) A suspended particulate matter distribution against a hydrographical background was studied at the oceanographic transect across the Equatorial Atlantic in the year 2000/

(1) L3 I think you mean: in the entire water column.
(2) Exactly. The text was changed.
(3) An area of abnormal high suspended matter volume concentrations was found above the Sierra Leone Rise in the entire water column (eastern part of the transect).

(1) L5 I suggest: This occurs within the Northwest African upwelling area …
(2) Corrected
(3) This occurs within the Northwest African upwelling area, where the plankton exposed to the Saharan dust abundance form a significant number of aggregates, which are lately transported equatorward via Canary Current.

(1)L5-7 "This process is located within the Northwest African upwelling area, since the plankton exposed to the Saharan dust abundance form a significant number of aggregates recently

transported equatorward via Canary Current." I do not understand the connection between the first part and the second part of this sentence. Please make clear and modify wording.

(2) I am sorry, the was sort of Frankenstein sentence. Corrections were made.

(3) This occurs within the Northwest African upwelling area, where the plankton exposed to the Saharan dust abundance form a significant number of aggregates, which are lately transported equatorward via Canary Current.

(1) L10-11 "The grain size of particles has a polymodal distribution with 2–4 μm and 8–13 μm modes." It that only for the AABW or for all particles? Please make clear."

(2) This was for the entire research.

(3) The grain size of particles along the entire transect has a polymodal distribution with 2--4 μm and 8–13 μm modes.

(1) L11-12 Idem ditto

(2) Corrected

(3) The registered in some parts of the transect rise in percentage of the 7--21 μm-sized particles suggests the presence of the well-known coarse mode (20--60 μm) formed by aggregation of transparent exopolymer particles (mucus).

(1) L14-15 delete: due to the fact of its importance (this is trivial)

(2) Corrected

(3) Numerous studies have focused on characterizing the suspended particulate matter (SPM) during the past half a century.

(1) L16 delete: Therefore,

(2) Done

(3)

(1) L23 GEOSECS is Geochemical Ocean Sections Study

(2) Corrected

(3) The Geochemical Ocean Sections Study (GEOSECS) introduced an early global description of suspended particle distri25bution in the ocean, using mostly data on SPM collected via filtration (Craig and Turekian, 1976).

(1) L23 an early GLOBAL description of …

(2) Corrected

(3) The Geochemical Ocean Sections Study (GEOSECS) introduced an early global description of suspended particle distri25bution in the ocean, using mostly data on SPM collected via filtration (Craig and Turekian, 1976).

(1) L25 … in Brewer et al. (1976) at … (format) Please also check at other places in the manuscript.

(2) Corrected. I am sorry, I suppose it is my insufficient English knowledge

(3) The three-layer model of the SMP concentration distribution was also successively described in Brewer et al. (1976) at a transect through the western Atlantic Ocean: high concentrations were found in surface and in rapidly moving bottom waters, while low concentrations were observed in the mid-water regions of the sub-tropical gyres.

(1) L29 This does not need to be in parentheses.

(2) Corrected

(3) The SPM plays a crucial role in both regulation of sea water composition and material transport throughout the entire 30water column, experiencing various processes e.g. dissolution, decomposition, disaggregation, aggregation, etc. (Gardner et al., 1985a).

(1) L42 … have been published in far less detail …
(2) Corrected
(3) The SPM data, on the contrary, have been published in far less detail (Sivkov et al., 2001) due to the fact that not all the peculiarities were in accordance with general knowledge (Biscaye and Eittreim, 1977), namely, high SPM concentrations in the clear water minimum layer in the open ocean of the Eastern Atlantic, which are not INL.

(1) L42-43 "due to the fact that not all the peculiarities were in accordance with general knowledge" This sentence makes the reader curious. Please give some example of this.
(2) We were referring to the vertical anomaly.
(3) The SPM data, on the contrary, have been published in far less detail (Sivkov et al., 2001) due to the fact that not all the peculiarities were in accordance with general knowledge (Biscaye and Eittreim, 1977), namely, high SPM concentrations in the clear water minimum layer in the open ocean in the Eastern Atlantic, which are not INL.

(1) L43-44 "So far, the ideas have been formed to explain the features of the SPM distribution obtained on the transect." It is not clear to me what this sentence wants to convey. Please modify.
(2) Corrected
(3) Recent studies of the SPM distribution and evolution allowed us to explain the features of the SPM distribution obtained on the transect.

(1) L47 "8th cruise of the R/V Akademik Ioffe" Is there any more info on this cruise that can be cited? Cruise report or report/publication in which the cruise is described?
(2) There were two publications, that were cited in the Line 56. The cruise report was only in Russian and was not published publicly (sorry for the word game). These articles were also referred in Introduction chapter.

(1) L48 "just before the start of the main transect" instead of "just before the transect conduction"
(2) Corrected
(3) Additional surface samples of SPM were taken in the Northwest African upwelling zone (Cap Blanc area) just before the start of the main transect (10–11 July 2000).

(1) L49 Please define "Ioffe-2000" in the previous sentence
(2) Corrected
(3) A sublatitudinal transect (13–28 July 2000) between the continental slopes of Guinea and Brazil with 61 closely spaced stations was performed during the 8th cruise of the R/V Akademik Ioffe (Fig. 1) (Ioffe-2000).

(1) L49 "the hydrological complex" This is not a common expression. Do you mean the CTD with rosette sampler? Please explain and modify
(2) We meant this.
(3) Sampling at the Ioffe-2000 transect was carried out using the CTD with rosette sampler, equipped with 5 L Niskin bottles, while the additional surface sampling was carried out with a use of plastic bucket.

(1) L58 Salinity is dimensionless, psu should not be used. Please indicate whether salinity was measured on the practical salinity scale.
(2) Corrected.
(3) (accuracy of measurements was 0.002°C for temperature, 0.002 for salinity on the practical salinity scale).

(1) Section 2 Please give the precision and accuracy of SPM concentration. Please add the size of the samples for determining SPM concentration.
(2) Sample volume was 0.5 ml, absolute measurement error for Coulter Counter is 6%.
(3) The SPM volume concentration and particle size distribution were determined by the conductometric method via the Coulter counter (Zbi model) for 407 samples from the Ioffe-2000 transect and 13 samples from the Cap Blanc area (Supplement 1) (0.5 ml sample). The Coulter counter calibration was carried out using Coulter Electronics standard methodology using 5.96 μm diameter latex particles. The 70 μm aperture was used, which ensured counting of particles in the size range of 1.8–20.7 μm. Volume suspended solids concentration and size distribution were calculated based on the assumption of particle sphericity. The absolute measurement error for Coulter Counter is 6% (Carder et al., 1974).

(1) L110 define NICC
(2) Corrected
(3) Northern Intermediate Counter Current

(1) L123 "The maximum of one of the CFCs (F11 — trichlorofluoromethane) represents LNADW" This is not quite clear. I suggest something like: The highest values of one of these CFCs (F11 — trichlorofluoromethane) are found in the LNADW.
(2) Corrected. Thank you!
(3) The highest values of one of these chlorofluorocarbonss (F11 — trichlorofluoromethane) are found in lower North Atlantic Deep Water (Rhein et al., 1998).

(1) L130 through (typo)
(2) Corrected
(3)

(1) L131 "(water with °4 > 45.90 kg m-3)" This is not clear. Do you mean water with temperature of 4°C?
(2) No, this is typo. The correct form is "(water with $\sigma_4 > 45.90$ kg m$^{-3}$)"
(3) The velocity field of Antarctic Bottom Water (water with $\sigma_4 > 45.90$ kg m$^{-3}$) is influenced by the overlying Deep Western Boundary Current (Rhein et al., 1998) (see Fig. 3).

(1) L132 "The AABW consists of old deep-water masses (Reid, 1994) with negligible F11 concentrations" This is a very old reference for such a contention about transient tracers. I could imagine that between 1994 and now the F11 concentration has increased. Please give more recent reference and check the contention. Actually, at L139 you mention the rising F11 in AABW.
(2) The rise of the F11 in AABW, mentioned at L139 is due to AABW contact with NADW. I've found the newer reference, but it is still not the modern data due to the fact that our article is based on the data from 2001.
(3) The Antarctic Bottom Water is older than North Atlantic Deep Water and has low F11 concentrations (Touratier et al., 2005) or even negligible in the Western Atlantic (Reid, 1994).

(1) L138 (> 0.07 pmol kg-1) (format of hyphen) Hyphen superscript should also be done in concentrations at other places in the text.

(2) Corrected

(3) The high level of F11 ($> 0.07$ pmol kg$^{-1}$) from lower North Atlantic Deep Water affects the part of Antarctic Bottom Water closest to the Ceara Rise, thus this Antarctic Bottom Water branch also experiences a rise in F11 content in comparison with its far eastern branch.

(1) Caption of Fig. 3 "1— sampling points, 2 — northward current, 3 — southward current, 4 — eastward current" Because it is not clear what the numbers stand for, I suggest to add in parentheses: (numbers indicated in panel bottom left)

(2) Corrected

(3) Distribution of the SPM volume concentration (ppm) on the Ioffe-2000 transect, done with Ocean Data View (Schlitzer, 2018). Numbers indicated in panel bottom left: 1 — sampling points, 2 — northward current, 3 — southward current, 4 — eastward current. Isolines are in geometric progression, for water mass abbreviations see text. Bottom topography from Sokov et al. (2002). The interpolation was done via the DIVA gridding (Barth et al., 2010) with 25 permille X-scale length and 65 permille Y-scale length.

(1) L161 delete "caused by primary production". One sentence later you explain that not only PP causes this but more factors.

(2) Corrected

(3) Accumulation in the surface layer may be the existence of the pycnocline at the bottom of the surface layer, which slows the sedimentation.

(1) L164 Please add: … accumulation in the surface layer may be the existence of the pycnocline at the bottom of the surface layer, which slows the sedimentation …

(2) Corrected.

(3) Accumulation in the surface layer may be the existence of the pycnocline at the bottom of the surface layer, which slows the sedimentation.

(1) L165 I suggest: "The "sedimentation barrier" can be seen as a visible pycnocline …

(2) Corrected

(3) The "sedimentation barrier" can be seen as a visible pycnocline at depths of 50–150 m to the west of the 30° W and 0–100 m to the east of this point on the Ioffe-2000 transect (Sarafanov et al., 2007).

(1) L169 calcium carbonate (CaCO3) … (note subscript for 3)

(2) Corrected

(3) The source of the vast majority of particles in the upper open ocean waters is internal and consists basically of particulate organic matter (POM), $CaCO_3$, and biogenic silica.

(1) L174/175 Please show the geographical names used here in the figures (map and possibly section plot)

(2) Guinea Marginal Plateau was added to Figure 1

(3)

(1) P8 and 9 The discussion of the SPM results makes frequent use of the hydrographic data. As the hydrographic results are not part of this manuscript, it is for the reader hard to understand. Please add a figure of the hydrography of the IOFFE-2000 section (potential temperature, salinity or density).
(2) We managed to find the initial temperature data, so the Figure 3 matched
(3)

[Figure]

Figure 3. Distribution of the potential temperature (°C) on the Ioffe-2000 transect (after (Sokov et al., 2002), done with Ocean Data View (Schlitzer, 2018). Numbers indicated in panel bottom left. 1 — northward current, 2 — southward current. For water mass abbreviations see text. Bottom topography from Sokov et al. (2002). The interpolation was done via the DIVA gridding (Barth et al., 2010) with 25 permille X-scale length and 25 permille Y-scale length. Dashed magenta line indicates $\sigma_4 > 45.90$ kg m$^{-3}$.

(1) L213 "at various mid-depths" I think this is not clear and well-defined. Please modify
(2) Corrected
(3) Clear water is referred to the minimum in the SPM concentration that is commonly located above the BNL (Eittreim and Ewing, 1974; Biscaye and Eittreim, 1977). Добавить мид-депт и под поверхностью

(1) L218 " … so the SPM concentrations were low (0.01–0.03 ppm)." Is this lower than usual in the clear water layer? Please specify
(2) It is low regarding our transect
(3) However, the intermediate waters on the Ioffe-2000 transect were not influenced by the lateral

source too strongly, so the SPM concentrations in the clear water were the lowest of the all transect (0.01–0.03 ppm).

(1) L229-230 "The water exchange in the deep layers of Eastern Atlantic …" What kind of exchange is meant here? Between which water masses?
(2) This was about the northward flow of AABW. Text was corrected
(3) The AABW transport in the deep layers of Eastern Atlantic is almost negligible in comparison with the intense northward flow of AABW to the east of the Ceara Rise in the western basin, that was noted during the Ioffe-2000 transect (Sarafanov et al., 2007).

(1) L236 above the Sierra Leone Rise (Fig. 3) (refer the Fig. 3)
(2) Corrected
(3) However, abnormally high SPM concentrations for these depths were noted above the Sierra Leone Rise (Fig. 4).

(1) L239 We explain this …
(2) Corrected
(3) We explain this "vertical anomaly" by using the ballast hypothesis (Armstrong et al., 2001; Louis et al., 2017), which is based on the correlation between the fluxes of particulate organic carbon (POC) and minerals in the deep ocean.

(1) Caption Fig. 5: Please indicate that this is a small deep part with more detail of the main contour plot in Fig. 3
(2) Corrected
(3) The SPM volume concentration at the western part of the Ioffe-2000 transect (magnified part with more detail of the Figure 3)

(1) L269-270 "an increase in the SPM concentrations" What increase do you mean? Compared to what? Or do you mean high values? With the one time section of SPM one cannot determine a difference, or increase.
(2) This is English problems again. We meant higher values in comparison with the Eastern Atlantic.
(3) The vast area to the west of the Mid-Atlantic Ridge within the Ioffe-2000 transect show higher (in comparison with the Eastern Atlantic) values of SPM concentrations, caused by Deep Western Boundary Current.

(1) L278-279 verb missing in sentence
(2) Corrected
(3) The deep cyclonic cell in the Guiana basin (McCartney, 1993; Arhan et al., 1998) at NADW depth is located near the Ceara Rise western slope and west of the MAR between 42°W and 36–37° W.

(1) L289 I do not understand the notation after AABW. Please explain or modify
(2) Sorry, this is the same typo as in L131 $\sigma_4 > 45.90$ kg m$^{-3}$
(3) The general northward pathway of Antarctic Bottom Water ($\sigma 4 > 45.90$ kg m$^{-3}$) across the Ioffe-2000 transect is located in the Para Abyssal Plain bottom layer.

(1) L300 Spell out m.a.b.
(2) Corrected
(3) The extremely high SPM concentration (0.36 ppm) was noted in bottom layer above the Ceara Rise (AI-903, depth 3678 m, 18 meters above bottom).

(1) L308 … were also noted above the MAR – Please refer to figure
(2) Corrected
(3) High levels of the SPM concentration were also noted above the Mid-Atlantic Ridge (Fig. 4).

(1) L309 change to: of NADW rich in the SPM
(2) Corrected
(3) There is the northward recirculation of North Atlantic Deep Water rich in the SPM (Sarafanov et al., 2007), which was noted both sides of the Mid-Atlantic Ridge at 36–40° W and 29–32° W.

(1) L322 "The overall cumulative number distribution for Ioffe-2000 transect …" Is this the mean of all stations of the Ioffe-2000 transect?
(2) Yes, it is

(1) L324-325 "The explanation for volume-size distributions was generally based on the results of the HEBBLE project in abyssal waters (the Nova Scotia Rise, 1977–1984)" It is not clear to me what you mean. Please clarify
(2) It is not clear indeed, I apologize.
(3) One of the first detail analysis of the SPM volume-size distributions was conducted during the HEBBLE project in abyssal waters…

(1) L328-329 "during the Ioffe-2000 cruise there was a possibility to count the fine part of the coarse mode only" This is not clear, I think you mean: … during the Ioffe-2000 cruise possibly only the fine part of the coarse mode was counted. Please change if needed
(2) I tried to make this sentence sound clearer
(3) Since the Coulter Counter aperture-tube truncation during the Ioffe-2000 was 70 μm, it was impossible to count particles up to 32 μm, so only the "left" part of the coarse mode is visible on Ioffe-2000 data (Fig. 9).

(1) L329 elevated instead of increased
(2) Corrected
(3) The volume-size distributions display: a) an elevated proportion of the 7–21 μm fraction, suggesting the coarse mode, b) a fine-mode with the peak size ranges about 2–4 μm, and c) an intermediate mode at 8–13 μm, that was not mentioned in the HEBBLE project.

(1) L332 "The most representative volume-size distributions in our study are shown in Figure 8." It is fine to show this figure with typical distributions. However, it does not become clear how many stations have such distributions and how many stations have different ones, i.e., no (semi-) quantitative info is given on the full transect. Would it be possible to give more such information?
(2) We changed both the text and the figure (now it is Figure 9) to make this part more clear. Added numbers of samples, while the figure became more simple — there was two graphs with intermediate mode. We personally find this mode interesting, but it was not so frequent to be represented in the Figure. We also have changed samples at the Figure — used ones, that were mentioned in the text for better clarity. The bottom maximum one and one from the vertical anomaly.
(3) The aggregation rate of 1.5–8 μm particles in the upper ocean depends on filtration by zooplankton. The fine mode in our data occurs within 3–8μm and in mostly in samples with low concentration of about 0.01 ppm (188 samples in total).

The "left" part of the coarse mode was noticed within the "vertical anomaly", as well as in the BNLs (88 samples in total). The intermediate mode (8–13 µm) was dominant in surface and intermediate waters (50–3000 m depth) in the samples with high SPM levels (0.3 ppm) (25 samples in total).

[Figure]

Figure 9. The typical examples of the bimodal volume-size SPM distribution. X-axis is particle diameter, Y-axis is probability density function. Green solid line — dominance of aggregates (station AI-903, 3678 m); dashed green line — intermediate mode, caused mostly by phytoplankton, detritus and transparent exopolymer particles (station AI-887, 3717 m). Size intervals are marked with color.

(1) L360, 367, 373 Use: First, Second and Third for clarity
(2) Corrected

(1) L365-366 "Our studies results agree with the fact that higher aggregate numbers and higher sinking velocities may increase the carbon export to deep ocean layers." Please give more detail where exactly you base this on.
(2) I paraphrased
(3) The vertical SPM concentration anomaly, revealed in our study, agrees with the fact that higher aggregate numbers and higher sinking velocities may increase the sediment transport to the bottom and since the biogenic aggregates are rich in carbon, the carbon export to deep ocean layers will also increase.

(1) L366-367 "Changes in dust sedimentation may bring about drastic changes in the global ocean due to the boost that trace nutrients from dust may cause for primary production." I think this does not follow from this work and should be deleted or needs a reference.
Agree. The sentence was deleted.

(1) L388 delete: 380
(2) Corrected

(1) References
L399 volume and page numbers missing
L444 editor of book missing
L459 Any more info on the report?
L461 editor of book missing
L467 The second author is called: Rutgers van der Loeff, M.
L475 Full reference: 117, C07002, doi:10.1029/2011JC007627.
L486 pages missing

L521 volume number missing
L541 full reference: 112, C12023, …
(2) All issues corrected

(1) Supplement
Please name the table and give a short description what is shown here.
(2) Corrected